# Vaterite/Fucoidan Hybrid Microparticles: Fabrication, Loading of Lactoferrin, Structural Characteristics and Functional Properties

**DOI:** 10.3390/md23110428

**Published:** 2025-11-05

**Authors:** Daniil V. Mosievich, Nadezhda G. Balabushevich, Pavel I. Mishin, Lyubov Y. Filatova, Marina A. Murina, Olga V. Pobeguts, Maria A. Galyamina, Ekaterina A. Obraztsova, Daria V. Grigorieva, Irina V. Gorudko, Alexey V. Sokolov, Ekaterina V. Shmeleva, Oleg M. Panasenko, Elena V. Mikhalchik

**Affiliations:** 1Department of Chemical Enzymology, Faculty of Chemistry, Lomonosov Moscow State University, 1 Leninskiye Gory, 119991 Moscow, Russia; dankir98@gmail.com (D.V.M.); nbalab2008@gmail.com (N.G.B.); pmishin2005@gmail.com (P.I.M.); luboff.filatova@gmail.com (L.Y.F.); 2Lopukhin Federal Research and Clinical Center of Physical-Chemical Medicine, 1a Malaya Pirogovskaya str., 119435 Moscow, Russia; marina_murina@mail.ru (M.A.M.); nikitishena@mail.ru (O.V.P.); mrogova@gmail.com (M.A.G.); e.a.obraztsova@gmail.com (E.A.O.); ekmos.vk@gmail.com (E.V.S.); o-panas@mail.ru (O.M.P.); 3Department of Biophysics, Belarusian State University, 4 Nezavisimosti Ave., 220030 Minsk, Belarus; dargr@tut.by (D.V.G.); irinagorudko@gmail.com (I.V.G.); 4Smorodintsev Research Institute of Influenza, 15/17 Prof. Popova Str., 197376 St. Petersburg, Russia; biochemsokolov@gmail.com

**Keywords:** vaterite, fucoidan, lactoferrin, anticoagulant properties, mucoadhesive properties, pH sensitivity, prolonged release, antibacterial properties, *B. subtilis*

## Abstract

Fucoidan is of considerable interest for the development of drug carriers. The inclusion of fucoidan allows calcium carbonate microparticles in the form of vaterite to acquire new properties, enabling their use in the immobilization of protein preparations. In this work, we investigated the properties of hybrid vaterite microparticles with fucoidan from *Fucus vesiculosus* obtained by co-precipitation and loaded with recombinant human lactoferrin from goats. The hybrid microparticles had a smaller diameter (3–4 µm), larger surface area (35–36 m^2^g^−1^), smaller pore size (5–10 nm average), and more negative ζ-potential (−(11–13) mV) than the control vaterite microparticles. The incorporation of lactoferrin into the microparticles by co-precipitation in complex with fucoidan was greater than when the protein was adsorbed onto the hybrid microparticles. Microparticles with fucoidan and lactoferrin were stable in acidic environments, released both components over a prolonged period at pH 7.4, and possessed mucoadhesive properties and anticoagulant activity. The antibacterial properties of hybrid microparticles with fucoidan and lactoferrin against *Bacillus subtilis* were characterized. Microparticles of vaterite with fucoidan can serve as a platform for the microfabrication of effective means of delivering therapeutic proteins.

## 1. Introduction

Recently, special attention has been paid to the development of complex means of delivering protein drugs using the natural polysaccharide fucoidan, especially with the use of modern technological approaches for obtaining nano- and microparticles [1,2,3,4,5,6,7].

Fucoidan belongs to a group of sulphated polysaccharides rich in L-fucose (Figure 1) but also contains other sugars, such as xylose, arabinose, rhamnose, glucose, galactose, and uronic acid. The main source of fucoidan is brown seaweed. The molecular weight of fucoidan (10–2000 kDa), its chemical structure, and sulfo group content vary depending on the type of algae, growing environment, harvesting season, and extraction method used [8,9]. Various biological activities of fucoidan have been described, including its anticoagulant, antioxidant, antitumour, antiviral, immunomodulatory, and anti-inflammatory properties [7,10].

Fucoidans from *Undaria pinnatifida* and *Fucus vesiculosus* are approved by the USA Food and Drug Administration (FDA) as generally recognized safe (GRAS) food ingredients at doses up to 250 mg day-1 [5]. Preparations containing *Fucus vesiculosus* are registered in a number of countries in Europe and Asia, including Russia [11].

Fucoidan is non-toxic, biodegradable, and biocompatible [5,12]. Being negatively charged, it forms complexes with positively charged molecules, which is why polyelectrolyte complexation is most often used to obtain various carriers using fucoidan [13,14,15,16,17,18]. Other common methods for obtaining biopolymer-based preparations are simple coacervation, ionotropic gelation, spray drying, emulsification, etc. [19]; additionally, nanoparticles [12,20], microparticles [12], films [21], hydrogels [20], and liposomes [22] have been obtained using fucoidan. Fucoidan-based particles have been studied as carriers for various drugs [23,24,25,26,27,28,29,30]. Furthermore, fucoidan was used not only as an excipient responsible for drug delivery, but also as a substance with an independent therapeutic effect [31,32] providing both protection of loaded substances and an increase in their effectiveness. 

It was shown that due to the negatively charged sulfate groups (pKa 1.0–2.5), good water-solubility and high charge density over a wide pH range, fucoidan is capable of forming stable complexes with positively charged proteins [13,14,15,16,17,18]. Such complexes can be soluble or insoluble depending on the fucoidan/protein ratio. When the pH decreases, the complexes break down due to the weakening of electrostatic interactions.

For 20 years, calcium carbonate has been actively studied for the manufacturing of inexpensive drug delivery systems [33,34,35,36,37,38,39]. Calcium carbonate preparations are already used in the food industry and are approved by the FDA in the USA as a medicinal product [34]. Particular attention is paid to vaterite polymorphic modification of calcium carbonate, which has a spherical shape and adjustable diameter (from 100 nm to tens of micrometers), high porosity, biocompatibility, and biosolubility [33,36]. The main disadvantage of vaterite is its thermodynamic instability, and it has been proposed to eliminate this disadvantage by introducing various biopolymers during the synthesis of hybrid vaterite particles [40,41,42]. The inclusion of natural anionic polysaccharides in calcium carbonate particles leads to changes in their structure, morphology, stability, and biological properties, as well as the efficiency of loading various biologically active substances [41,42,43]. Hybrid microparticles with a diameter of 3–4 μm were successfully prepared using fucoidan from *Fucus vesiculosus* and proved to be convenient matrices for the immobilization of cationic proteins [41]. In other studies, low-molecular-weight drugs were incorporated into calcium carbonate nanoparticles with fucoidan [44,45]. In the group of Guo Y. [44], hybrid vaterite nanoparticles with fucoidan with a diameter of 261 nm were formed by gas diffusion, into which the cytostatic drug methotrexate was successfully loaded. In the group of Liu Y. [45], calcium carbonate nanorods with fucoidan 600 nm long and 130 nm wide were obtained, into which the antitumour drug mitoxantrone hydrochloride was adsorbed, the release of which was pH-dependent.

Earlier, we studied the effect of fucoidan from *Fucus vesiculosus*, incorporated into the hybrid vaterite microparticles, on the adsorption of therapeutically important proteins (albumin, catalase, chymotrypsin) and the glycoprotein mucin, which is the main component of mucous membranes [41]. We have also shown that both vaterite and vaterite–fucoidan microparticles have no cytotoxic effects on human cells in model systems, and the lysis of erythrocytes and HT-29 cells by the CC and CCF in vitro did not exceed 3% [41].

In this work, lactoferrin was studied as the target protein for incorporation in particles with fucoidan since this protein is already used in the food industry and for the prevention and treatment of various diseases [46,47,48,49,50], including cancer [51].

Lactoferrin is a glycoprotein with a molecular weight of about 80 kDa, consists of a single polypeptide chain comprising 680–703 amino acid residues, and has a high affinity for iron [46,52]. The isoelectric point of lactoferrin is in the range of 8.0–8.5 [52], so the biopolymer is positively charged at physiological pH values [46].

Due to its strategic position on the surface of the mucous membrane, lactoferrin is one of the first lines of defense against microbial agents that enter the body mainly through the mucosa. The involvement of lactoferrin in apoptotic processes in cancer cells, its ability to modulate various immune system responses, and its activity against a wide range of pathogenic microorganisms, including respiratory viruses, have made lactoferrin a subject of broad interest for pharmaceutical research and the development of various oral delivery systems [50,53,54].

The loading of lactoferrin into vaterite microparticles by adsorption or co-precipitation, followed by the production of polyelectrolyte capsules by alternating adsorption of bovine serum albumin/tannic acid and dissolution of the CaCO_3_ matrix, has been described in [55]. Kiryukhin M. et al. [56] compared the adsorption of lactoferrin on vaterite microparticles and microparticles made of commercial natural calcium carbonate treated with phosphoric acid to increase the surface area. In [57], vaterite microparticles were grown in the presence of lactoferrin on hydrogel beads made of sodium alginate and gellan gum. 

In this work, to increase the functionality of vaterite microparticles as protein carriers, we modified their structure by co-precipitation with fucoidan. To increase stability and provide additional protection of lactoferrin, it was loaded into vaterite microparticles in the form of a polyelectrolyte complex with fucoidan.

Scanning electron microscopy (SEM), energy-dispersive X-ray spectroscopy (EDS), fluorescence microscopy, spectroscopy, X-ray diffraction analysis (XRD), thermogravimetric analysis (TGA), electrophoretic light scattering (ELS), low-temperature nitrogen adsorption–desorption, coagulometry, and microbiological methods were used to analyze the properties of hybrid vaterite-based microparticles, including their mucoadhesive and biological properties, and the components included. The aim of the work is to study the effect of fucoidan on the production, properties, and stability of multifunctional hybrid particles of vaterite with lactoferrin.

## 2. Results and Discussion 

### 2.1. Obtaining Hybrid Microparticles and Incorporating Fucoidan

Vaterite microparticles were obtained by spontaneous crystallization through rapid addition of sodium carbonate solution to calcium chloride solution in the presence of 0.05 M Tris buffer to lower the pH in the reaction medium; then, the microparticles were washed and freeze-dried (Figure 2).

No additives were introduced during the synthesis of control vaterite microparticles (CC). To obtain hybrid microparticles with fucoidan (CCF), the polysaccharide fucoidan (F) was pre-mixed with a calcium chloride solution.

To determine the optimal strategy for incorporating lactoferrin (L, 80 kDa; pI 8.0–8.5) into microparticles, two loading methods were compared: co-precipitation and adsorption (Figure 2). As described below in the Section 3, lactoferrin adsorption was performed on dried CCF particles to obtain CCF-L microparticles. For co-precipitation, lactoferrin and fucoidan were pre-incubated in a 1:1 mass ratio to form the FL polyelectrolyte complex, the formation of which we had previously confirmed [58], and then quickly mixed with calcium chloride and sodium carbonate solutions to obtain CCFL microparticles.

The content of lactoferrin co-precipitated as a complex with fucoidan in CCFL microparticles was 26 ± 15 mg g^−1^, which was higher than its adsorption in CCF-L microparticles (16 ± 2 mg g^−1^). Therefore, the approach with the preliminary formation of a polyelectrolyte complex and its co-precipitation was further used to obtain hybrid particles with fucoidan and lactoferrin.

### 2.2. Physicochemical Properties of Microparticles 

The main physicochemical characteristics of both control CC and hybrid microparticles with fucoidan and lactoferrin are presented in Table 1.

As the SEM data indicated, all hybrid microparticles, like the control CC, were spherical and porous (Figure 3). The diameter of the microparticles decreased slightly during the co-precipitation of fucoidan and lactoferrin, as can be seen from the particle size distribution diagrams (Figure 3A–C). CCFL microparticles had the smallest diameter (~3 μm). The sulfur in CCF and CCFL microparticles was detected (Table 1) using EDS spectra (see Appendix A).

The TGA method and the Dubois method (see description and Appendix A) were used to quantitatively determine the fucoidan content in hybrid microparticles CCF (Table 1). Based on the change in the mass of the preparations in the temperature range of 171–559 °C, which occurred due to the decomposition of the polysaccharide, the fucoidan content in the CCF microparticles was 4.7% (Table 1). Estimations made using the Dubois method with a calibration curve for fucoidan (Appendix A) showed that the polysaccharide content in the CCF and CCFL microparticles was approximately 10% (Table 1).

The lactoferrin content in CCFL microparticles after dissolution of calcium carbonate with hydrochloric acid was determined by the Lowry method and amounted to 2.7% (see description and Appendix A) as represented in Table 1.

According to XRD data (Appendix A), the proportion of vaterite in microparticles with CCF was higher than in CC, 99% and 97%, respectively. The lower calcite content indicated the stabilization of microparticles by polysaccharide.

Using the Brunauer–Emmett–Teller (BET) low-temperature nitrogen adsorption/desorption method, isotherms were obtained (Figure 4A–C) and the surface area, specific volume, and average pore size of the microparticles were determined (Table 1, Figure 4D). The isotherms of all vaterite microparticles were of type IV according IUPAC classification and had a hysteresis loop, indicating capillary condensation in mesopores. The form of hysteresis loop for all microparticles corresponded to the mixed type A and E (cylindrical or blind pores). The hysteresis loop closed in the process of desorption in all isotherms earlier than the relative pressure achieved the value of 0.3, which could be explained by the absence of micropores. Compared to the control CC, the hybrid CCF and CCFL microparticles had almost three times larger surface area and smaller pore diameter (Table 1). The CCFL microparticles had particularly small pore sizes (5.4 nm) and the narrowest pore size distribution (Figure 4D).

Thus, hybrid microparticles with fucoidan (CCF) and with fucoidan and lactoferrin (CCFL) represented a vaterite modification of calcium carbonate, contained biopolymers, had a smaller diameter and smaller pore size, and a high surface area compared to the control microparticles (CC). The similar effects were observed when lysozyme [36] or various polysaccharides (pectin, heparin, chondroitin sulfate, fucoidan) [41] were co-precipitated with CaCO_3_ yielding hybrid vaterite microparticles. It can be proposed that biopolymers increase the viscosity of the reaction medium which could influence the rate of Ca^2+^ diffusion and decrease the rate of crystals growth. Although lactoferrin and fucoidan studied in the present work can form a polyelectrolyte complex influencing the fucoidan structure according data of FTIR [58], it is difficult to register such effects in hybrid microparticles. As it was shown for mucin–vaterite crystals, no appearance of characteristic bands of mucin could be detected that might be related to low signal to noise ratio of characteristic amide bonds to the CaCO_3_ background that is below detection limit [42].

Further, we investigated the properties of microparticles that are important for their functional activity, such as mucoadhesiveness, the ability to prolong the release of the included components and preserve their biological activity.

### 2.3. Mucosal Adhesive Properties of Microparticles

For mucosal delivery, it is necessary that the carrier particles are able to attach and remain on the mucous membrane, the main component of which is the high-molecular-weight glycoprotein mucin, and release the target drugs in a prolonged manner. Previously [43], the mucoadhesive properties of vaterite microparticles CC were demonstrated by mucin adsorption, so it is important to preserve them in hybrid microparticles.

When studying the binding of FITC-labeled mucin (MF) to particles using fluorescence microscopy, the adsorption of glycoprotein on hybrid microparticles CCF and CCFL was visualized (Figure 5).

Fluorescence spectroscopy of supernatant solutions separated after binding with mucin-FITC microparticles (Figure 6) revealed lower adsorption of high-molecular-weight glycoprotein (≈600 kDa) on CCFL microparticles with the smallest pore size.

Additionally, the sorption of FITC-mucin by microparticles was analyzed using flow cytometry (Figure 7).

The median FITC fluorescence intensity decreased in the order of microparticles CC > CCF > CCFL and was consistent with the results of adsorption and reduction in pore size (Table 1). The average hydrodynamic diameter of mucin in 1 mg mL^−1^ water solution was 40 or 250 nm with negative zeta potential of −15 mV for both fractions [43,59], which presumably corresponded to the aggregated mucin and to the single mucin molecules. It was shown that FITC-mucin can penetrate into the pores of CC but predominantly it is located on the edges of the particles. Penetration of FITC-mucin into the pores of CCF and CCFL can be hindered by smaller pore diameter of hybrid microparticles compared with CC. Earlier, we showed that adsorption of mucin by CCF microparticles was 2.3 times less than by CC [41], which supports our present results. So, it is more likely that FITC-mucin is located on the particle surface and the pore size does not influence its fluorescence.

### 2.4. Stability of Hybrid Microparticles and Release of Components

The study of the stability of the polymorphic modification of vaterite and the release of the carried substances from microparticles is a key characteristic of complex preparations based on calcium carbonate, which are promising for oral delivery [42]. The release of loaded drugs from vaterite microparticles is possible as a result of three processes: particle dissolution, diffusion from pores, and recrystallisation of the vaterite modification into thermally more stable calcite, which has a low capacity for large macromolecules [34]. For oral delivery to the intestine, particle stability and low drug release in acidic environments corresponding to gastric juice are important.

In order to predict the behavior of hybrid microparticles in various physiological environments, we studied how pH in the range from 2 to 8 affected the release of fucoidan and lactoferrin during 1 h of incubation in a universal buffer containing acetic, boric, and phosphoric acids (Figure 8A,B). It should be noted that the ζ-potential of the particles was negative independently of medium pH which can be explained by the presence of fucoidan on the particles surface.

The release of fucoidan (Figure 8A) from CCFL microparticles containing a protein–polysaccharide complex was lower than from CCF particles, especially at pH 2–4. The release of lactoferrin (Figure 8B) from CCFL microparticles at pH 3–6 was lower than at pH 2 and pH 7–8. Apparently, at pH below the isoelectric point (pI 8–8.5), lactoferrin is present in microparticles in the form of a complex with fucoidan, whose sulfate groups have a pK_a_ of 1–2.5. In acidic environments with a pH close to the pKa of fucoidan, as well as in environments with a pH close to the pI of the protein, partial destruction of the FL complex with the release of lactoferrin is possible, leading to its release from the CCFL particles. A similar effect of pH on protein release has been shown for complexes of fucoidan with serum proteins β-lactoglobulin [18], ovalbumin [17], and lactoferrin from cow’s milk [14].

The kinetics of the release of all components from hybrid microparticles were studied under conditions simulating the small intestine environment (Figure 8C,D), namely, by constant incubation of the preparations in PBS buffer with a pH of 7.4 for 24 h as de-scribed in Section 3.4. All hybrid microparticles with fucoidan remained negatively charged (according to ζ-potential data) and spherical (according to optical microscopy within 24 h). Hence, there was no transformation of vaterite to calcite with loss of fucoidan. It should be noted that even after 168 h incubation under the same conditions the particles retained their structure and stability.

The release of fucoidan (Figure 8C) from vaterite microparticles was rapid initially, apparently due to the partial dissolution of CaCO_3_ on the surface of the particles, and then slowed down due to diffusion and reached a plateau. The release of fucoidan from CCFL microparticles with the fucoidan–lactoferrin complex was slower and lower than that from CCF microparticles. The manner of lactoferrin release (Figure 8D) from CCFL microparticles was similar, and its total release did not exceed 5%.

Thus, hybrid microparticles CCF and CCFL were stable in acidic environments and released components over a prolonged period at pH 7.4. These properties make hybrid microparticles of vaterite with fucoidan and proteins promising for mucosal, including oral, delivery.

### 2.5. Anticoagulant Properties of Microparticles 

The anticoagulant properties of fucoidan and microparticles with fucoidan CCF were studied by determining the activated partial thromboplastin time (APTT) of blood plasma (Figure 9).

The control microparticles of vaterite CC did not exhibit anticoagulant properties. Microparticles with fucoidan (CCF) inhibited blood plasma coagulation more strongly than microparticles with fucoidan-lactoferrin complex(CCFL). This is due to the fact that the presence of fucoidan in a polyelectrolyte complex with lactoferrin reduces the manifestation of the anticoagulant properties of the sulfopolysaccharide [58]. The anticoagulant properties of fucoidan in particles may be associated with its presence on their surface and/or its release into solution individually or in the form of a complex. To distinguish between these effects, APTT was additionally analyzed in supernatant solutions separated after incubation of microparticles in a 0.9% NaCl solution (Figure 10).

With increasing the incubation time, the blood plasma coagulation time with supernatants increased slightly. Fucoidan on the surface of microparticles lost up to 25% of its activity. This indicates that the anticoagulant effects of microparticles are largely due to surface-bound sulfopolysaccharide.

### 2.6. Effects of Microparticles and Their Components on Bacteria Bacillus subtilis

Fucoidan can be considered as a potential prebiotic [60] while lactoferrin has antibacterial activity due to the binding of iron ions necessary for bacterial growth (bacteriostatic effect), bactericidal action due to its interaction with lipopolysaccharides and lipoteichoic acids of bacterial membranes or due to the formation of peptides that damage these membranes, and also due to proteolytic activity against the virulence factors of some bacteria [61]. The effects of lactoferrin on bacterial cultures depend on the type of microorganism, which is associated with the peculiarities of the structure of their cell membranes [62].

The object of our study was the bacterium *Bacillus subtilis RIK 1285*. *Bacillus subtilis* are widely used as probiotics, so assessing the effect of fucoidan on the growth of these bacteria is of practical interest.

These bacteria are sensitive to lactoferrin, which was important for analyzing the activity of lactoferrin in complex with fucoidan FL and in the composition of hybrid vaterite microparticles. According to measurements of absorption at 540 nm (Figure 11A) and total nucleic acid concentration (Figure 11B) in suspensions, bacterial growth increased in the presence of fucoidan but decreased in the presence of both lactoferrin and the fucoidan–lactoferrin complex.

When bacteria were grown in the presence of microparticles, there was a significant decrease in nucleic acid concentration in bacteria cultured with CCFL as compared to bacteria cultured in the presence of CCF (Figure 11C). This indicated that the antibacterial activity of lactoferrin co-precipitated in the form of a complex with fucoidan was retained in the microparticles.

Thus, fucoidan promoted the growth of *B. subtilis RIK 1285*, i.e., it is capable of performing the function of a potential prebiotic, while lactoferrin separately, in combination with fucoidan FL or as part of CCFL microparticles, retained its antibacterial effect.

## 3. Materials and Methods

### 3.1. Materials

Anhydrous calcium chloride, ≥93.0% (C1016), anhydrous sodium carbonate, ≥99.0% (S7795), fucoidan from *Fucus vesiculosus*, 30–100 kDa (F5631), and mucin from porcine stomach, Type III (M) (Sigma–Aldrich, Burlington, MA, USA) were used. Recombinant human lactoferrin from the milk of transgenic goats (preparation “Caprabel”, Belarus) was employed. All chemicals for buffer preparation were of at least laboratory grade and purchased from Sigma–Aldrich. Other chemicals were used without further purification. Double-distilled water was used in all experiments.

### 3.2. Preparation of Vaterite Microparticles

The synthesis of control vaterite microparticles (CC), hybrid microparticles with fucoidan (CCF), and complex hybrid microparticles with fucoidan and lactoferrin (CCFL) was performed according to the method developed by the group of N. Balabushevich N. [41]. In a beaker with a bottom diameter of 45 mm under magnetic stirring (200 rpm), the required volumes of solutions were sequentially added as specified in Table 2. For CCF microparticles, 3 mL of 1 M CaCl_2_, 7.5 mL 10 mg mL^−1^ of fucoidan in 0.1 M Tris buffer, and 1.5 mL water was stirred for 10 min followed by the addition of 3 mL of 1 M Na_2_CO_3_. Stirring was continued for 45 s; then, the suspension was left for 15 min without stirring for precipitation. CC microcrystals were fabricated following the same procedure without addition of polysaccharide into the CaCl_2_ solution. For CCFL microparticles, fucoidan and lactoferrin were pre-incubated for 10 min to form a complex. After synthesis, the microparticles were centrifuged at 1000 rpm for 1 min, washed twice with double-distilled water, frozen, lyophilized (freezing at −40 °C and drying for 24 h), and weighed.

For lactoferrin adsorption, 40 mg of microparticles was mixed with 1 mL of a 1 mg mL^−1^ protein solution in 0.05 M Tris-buffer (pH 7.0). The suspension was incubated for 30 min at 300 rpm, centrifuged at 2000 rpm for 2 min, and the pellet was washed twice with 0.05 M Tris-buffer (pH 7.0).

For both incorporation methods, the lactoferrin concentration in the supernatant and wash solutions was analyzed by Lowry’s method [63] (see description in Appendix A). The incorporation efficiency was determined based on the decrease in lactoferrin in the processing solutions, and its content in the dry samples was calculated. Fucoidan content was determined using the Dubois method [64].

### 3.3. Characterization of the Microcrystals

#### 3.3.1. Scanning Electron Microscopy

Scanning electron microscopy (SEM) was performed using a Zeiss DSM 40 microscope (Zeiss, Munich, Germany). Dry microparticle samples were deposited on silicon wafers. The accelerating voltage was set to 1–2 kV. ImageJ version 1.52k software was used for statistical analysis of 100 microparticles per sample to determine the average diameter (D_particle_).

Elemental composition was analyzed using energy-dispersive X-ray spectroscopy (EDS) during SEM measurements with an Oxford Instruments INCAx-act attachment. Single EDS spectra were obtained in regions of interest to calculate the relative content of chemical elements, and elemental distribution maps were generated to assess sample homogeneity. EDS data were recorded at higher accelerating voltages (10–15 kV).

#### 3.3.2. Low-Temperature Nitrogen Adsorption–Desorption

Nitrogen adsorption was performed using a Micromeritics ASAP-2020 analyzer (Micromeritics, Norcross, GA, USA). Samples were degassed at room temperature to 10^−3^ Pa prior to analysis. The isotherms were recorded at 196 °C as the dependences of the volume of the sorbed N_2_ (cm^3^g^−1^) on the relative pressure p/p_0_, where p_0_ is the pressure of saturated N_2_ vapor at −196 °C. Specific surface area (S) was determined using the Brunauer–Emmett–Teller (BET) method, pore volume (V) was assessed using the t-plot method, and pore size (D_pore_) was calculated using the Barrett–Joyner–Halenda method.

#### 3.3.3. Electrophoretic Light Scattering

Zeta potential of fucoidan, lactoferrin, and microparticles was measured using electrophoretic light scattering (ELS) on a Zetasizer Nano ZS (Nano ZS, Malvern, UK). Solutions of biopolymers (1 mg mL^−1^) and microparticle suspensions (0.1 mg mL^−1^) were analyzed.

#### 3.3.4. X-Ray Diffraction Analysis

X-ray diffraction (XRD) of vaterite microparticles was performed on a Miniflex 600 diffractometer (Rigaku, Tokyo, Japan) with a CuKα_1,_ _2_ source and a D/teX Ultra semiconductor detector (CuKα_1_ λ = 1.5418 Å) in reflection mode (2θ = 4–60°, step 0.02°, scan rate 5°/min). Phase composition (Appendix A) was determined using the following formula:$$\frac{I_{c}^{104}}{I_{c}^{110}}=7.691\times \;\frac{X_{c}}{X_{v}}$$
where 7.691 is the proportionality constant, $\frac{I_{c}^{104}}{I_{c}^{110}}$ is the intensity ratio of calcite (104) and vaterite (110) peaks, and *X_c_/X_v_*, is the molar ratio of calcite to vaterite in the samples.

#### 3.3.5. Thermogravimetric Analysis

Thermogravimetric analysis (TGA) was conducted on an SDT Q600 analyzer (Thermo Fisher Scientific, TA Instruments, Waltham, MA, USA). Samples were heated from 30 to 1000 °C at 10 °C min^−1^ under an air flow (100 mL min^−1^). For the processing and interpretation of TGA data for CCF, the TGA data for fucoidan from *F. vesiculosus* [65] was used.

#### 3.3.6. Coagulometry

Activated partial thromboplastin time (APTT) was measured using a coagulometer (APG4-03-Ph, EKMO, Moscow, Russia). Clotting time was determined mechanically by adding a metal ball, 40 μL plasma, 10 μL test solution, incubating at 37 °C for 1 min, adding 50 μL cephalin–kaolin mixture, incubating for 2 min, and then adding 50 μL of 0.025 M CaCl_2_.

#### 3.3.7. FITC-Mucin Adsorption on Microparticles

Fluorescein isothiocyanate-labeled mucin (FITC-mucin, FM) was prepared as described in [43]. For adsorption, equal volumes of microparticle suspension and FITC-mucin in 0.15 M NaCl were mixed to final concentrations of 10 mg mL^−1^ (particles) and 0.5 mg mL^−1^ (mucin). After 30 min incubation, samples were centrifuged (2000 rpm, 2 min), and supernatants were analyzed for unbound FITC-mucin fluorescence. Washed pellets were used for fluorescence microscopy. 

To study the fluorescence intensity of the particles, the sediment was resuspended in water to a final particle concentration of 1 mg mL^−1^. The resulting suspensions were then analyzed on a LongCyte flow cytometer (30 μL sample, flow rate 60 μL min^–1^). Forward scatter (FSC) and side scatter (SSC) signals were recorded, as well as four fluorescence channels (530/30, 577/25, 690/50, 780/60 nm), with an excitation laser wavelength of 488 nm. Singlets and doublets were distinguished using SSC-W and SSC-A graphs.

#### 3.3.8. Fluorescence Microscopy

FITC-mucin-adsorbed microparticles were examined using a Nikon Ni-E microscope (Nikon, Tokyo, Japan) with an FITC filter (excitation: 465–495 nm, emission: 512–555 nm).

### 3.4. Component Release Studies 

To study pH-dependent release, 10 mg microparticles were mixed with 250 μL universal buffer (0.02 M H_3_PO_4_, 0.02 M CH_3_COOH, 0.02 M H_3_BO_3_ + 1 M NaOH, pH 2–8) [66]. Suspensions were incubated at 37 °C (220 rpm, 1 h), centrifuged, and supernatants were analyzed for lactoferrin and fucoidan.

For release kinetics, five test-tubes each with 20 mg microparticles were prepared and microparticles were mixed with 1.5 mL PBS (pH 7.4). Then after 10 min, 1 h, 3 h, 6 h, 24 h, and 168 h incubation times, one probe was analyzed by light microscopy and the particles in other probes were separated by centrifugation (2000 rpm, 2 min) for further assay of lactoferrin content according to the Lowry method [65] and fucoidan content according to the Dubois method [66].

### 3.5. Evaluation of Biopolymer and Particle Effects on Bacillus subtilis Culture

#### 3.5.1. Bacterial Cultivation

The laboratory strain *B. subtilis RIK 1285* Marburg 168 derivative: *trpC2*, *lys1*, *aprE Δ3*, *nprR2*, *nprE18* from Takara Bio Inc was inoculated into M9 medium and grown at 37 °C with stirring at 180 rpm overnight. To obtain bacteria in the stationary growth phase, 180 μL of bacteria diluted 4 times with M9 medium and grown overnight was added to the wells of a 96-well plate. Then, either 20 μL of the test samples of lactoferrin, fucoidan, or their complex was pre-incubated for 30 min at 25 °C; a final concentration of each component of 0.3125 mg mL^−1^, or a suspension of microparticles to a final concentration of 3.125 mg mL^−1^, were then added. M9 medium was added to separate wells instead of bacteria, and the test samples were added in the same concentrations as in the wells with bacteria. The bacteria were grown for 22 h at 37 °C, after which the optical absorption of the suspensions at 540 nm was measured in the wells using an Ascent plate spectrophotometer.

#### 3.5.2. Determination of Nucleic Acid Concentration

To determine the concentration of nucleic acids in samples with bacteria, the Spirin method [67] was used. Hydrochloric acid solution (1 M) was added to the test sample to a volume ratio of 1:1 and mixed. The mixture was heated on a thermostat to 99 °C for 20 min. After cooling, the samples were centrifuged for 10 min at 10,000 rpm and the optical absorption in the supernatant was measured at 270 and 290 nm. The nucleic acid content as μg mL^−1^ was calculated using the following formula:C = (A_270_ − A_290_)/0.19 × 10.3 × 2,
where A_270_ and A_290_ are the optical absorption values at 270 and 290 nm; 0.19 is the specific extinction coefficient of nucleic acids; 10.3 is the conversion factor for phosphorus to nucleic acids, μg mL^−1^; and 2 is the dilution of the test sample.

### 3.6. Data Processing and Statistical Analysis

Calculations and statistical processing of the results were performed using Excel, Origin, and Statistica 12.0 programs. The results are presented as Mean ± SD, where Mean is arithmetic mean and SD is standard deviation. Statistical significance of differences between groups was assessed using Student’s *t*-test for independent variables and Mann–Whitney test. A difference was considered significant at *p* < 0.05.

## 4. Conclusions

In this study, hybrid vaterite microparticles were obtained and comprehensively investigated using natural sulfopolysaccharide fucoidan from *Fucus vesiculosus.* We also studied the suitability of these hybrid vaterite microparticles with fucoidan for the inclusion and release of a therapeutically important protein, recombinant human lactoferrin from the milk of transgenic goats.

Co-precipitation of the sulfopolysaccharide during the preparation of the drugs led to a change in the physicochemical properties of the hybrid vaterite microspheres with fucoidan; namely, a reduced diameter, increased surface area, reduced pore size, and a more negative ζ-potential compared to the control vaterite particles. These properties make the use of hybrid particles of vaterite with fucoidan promising for the inclusion of positively charged biologically active substances. The example of recombinant lactoferrin demonstrates the possibility of incorporating therapeutically important proteins into hybrid microparticles with fucoidan. The incorporation of protein into microparticles by co-precipitation in the form of a complex with fucoidan led to an increase in the protein content compared to adsorption on ready-made hybrid microparticles. The greater stability of hybrid microparticles of vaterite with fucoidan and lactoferrin in acidic environments and the prolonged release of fucoidan and lactoferrin under conditions modeling the human small intestine, as well as the preservation of the mucoadhesive properties of the microparticles, indicate the prospects for their use in oral delivery, including in the treatment of inflammatory bowel diseases [68].

The preservation of the functional properties of fucoidan incorporated into vaterite microparticles separately or in the form of a polyelectrolyte complex with lactoferrin was demonstrated in a study of anticoagulant properties.

Using the strain *B. subtilis RIK 1285*, it was found that fucoidan from Fucus vesiculosus promoted bacterial growth, i.e., fucoidan is capable of functioning as a potential prebiotic. Antibacterial properties were identified in recombinant lactoferrin from the milk of transgenic goats, a complex of fucoidan and lactoferrin, and hybrid microparticles with fucoidan and lactoferrin. This allows us to discuss the prospects for the use of fucoidan and lactoferrin in microparticles for regulation of intestinal microbiota.

## Figures and Tables

**Figure 1 marinedrugs-23-00428-f001:**
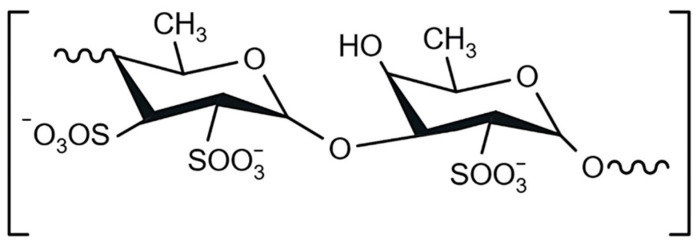
Chemical structure of fucoidan.

**Figure 2 marinedrugs-23-00428-f002:**
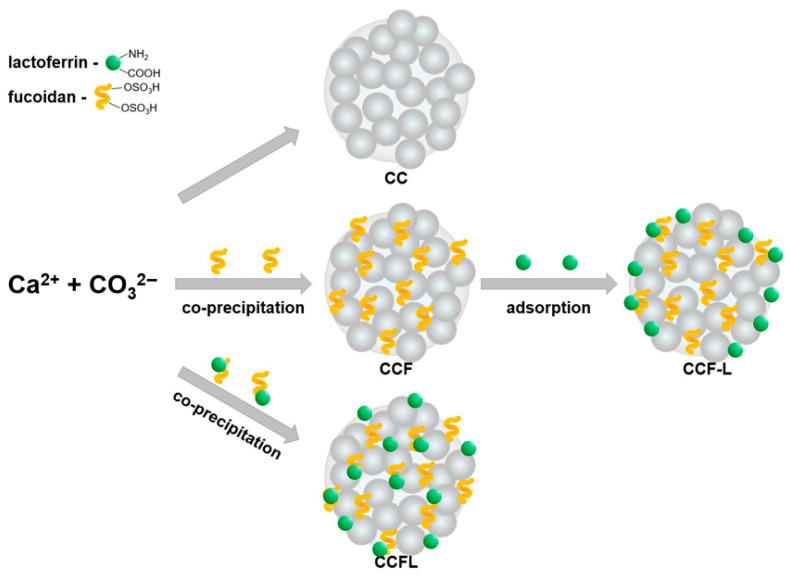
Scheme for obtaining control and hybrid vaterite microparticles with fucoidan and lactoferrin (CC, vaterite; F, fucoidan; L, lactoferrin; FL, complex).

**Figure 3 marinedrugs-23-00428-f003:**
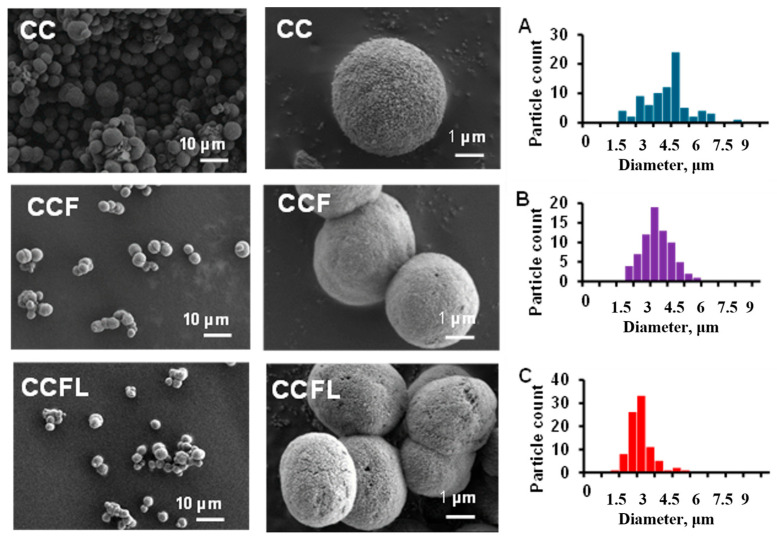
SEM and diameter distribution diagram of microparticles: control CC (**A**), hybrid with fucoidan CCF (**B**), and hybrid microparticles with fucoidan and lactoferrin CCFL (**C**). EDS spectra of the microparticles are represented in the Appendix A.

**Figure 4 marinedrugs-23-00428-f004:**
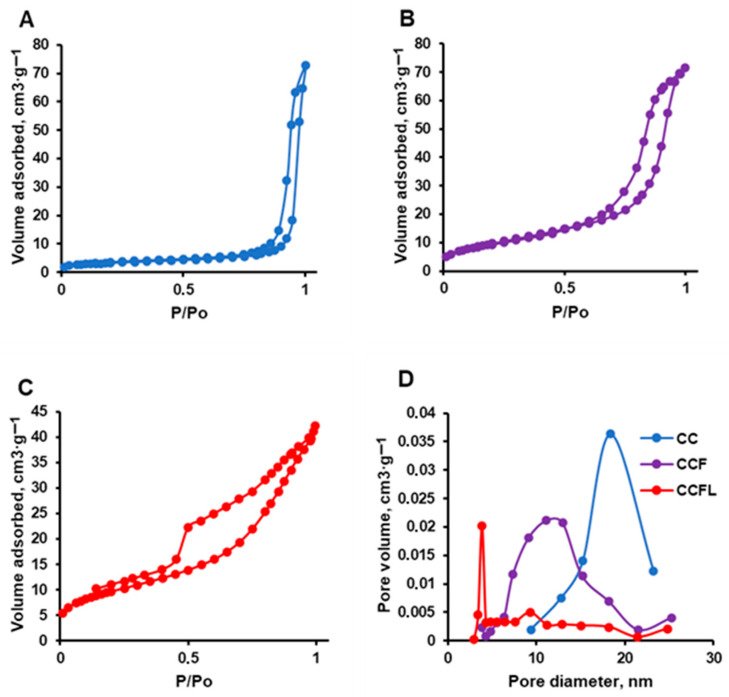
Nitrogen adsorption/desorption isotherms obtained by the BET method for CC (**A**), CCF (**B**), and CCFL (**C**) microparticles, and pore size distributions for these microparticles (**D**).

**Figure 5 marinedrugs-23-00428-f005:**
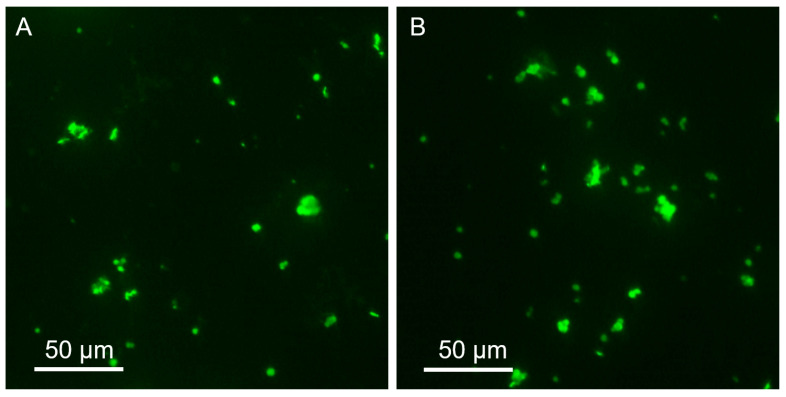
Fluorescence microscopy of microparticles treated with FITC-mucin: CCF-MF (**A**) and CCFL-MF (**B**).

**Figure 6 marinedrugs-23-00428-f006:**
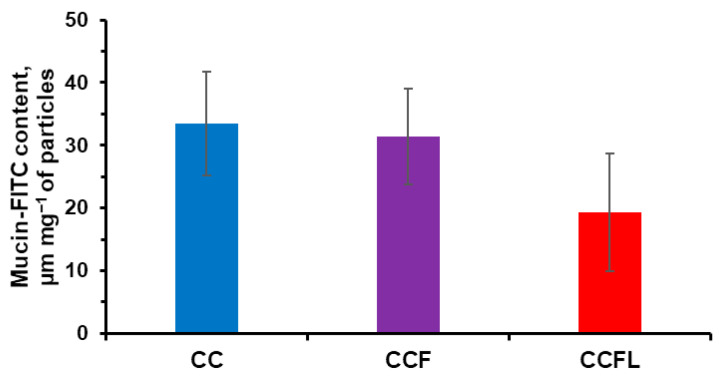
Fluorescence spectroscopy data on the binding of FITC-mucin by vaterite microparticles (based on the analysis of supernatant solutions).

**Figure 7 marinedrugs-23-00428-f007:**
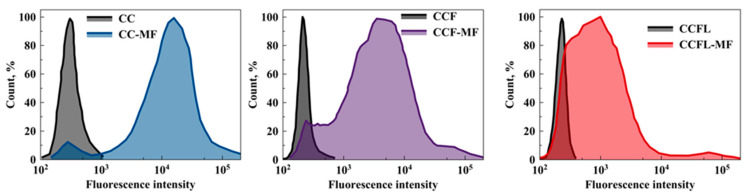
Diagrams of FITC fluorescence intensity distribution for CC, CCF and CCFL vaterite microparticles before and after FITC-mucin sorption.

**Figure 8 marinedrugs-23-00428-f008:**
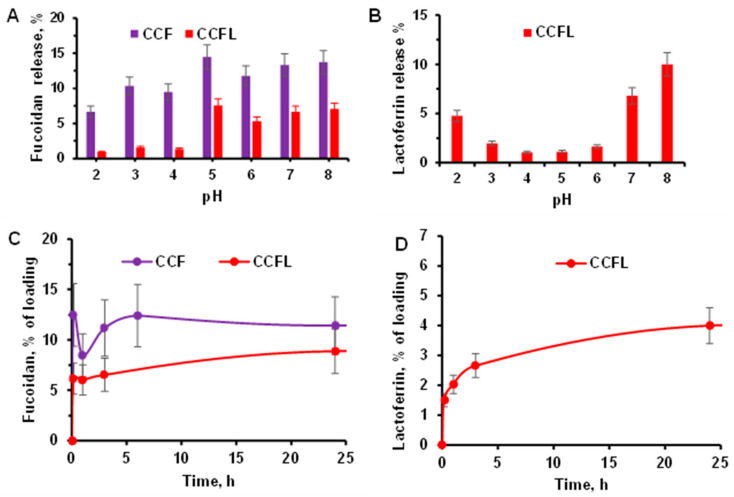
(**A**,**B**). Effect of pH on the release of fucoidan (**A**) and protein (**B**) from CCF and CCFL microparticles during incubation in a universal buffer. The effect is evaluated as a percentage of the amount fucoidan or lactoferrin included in the particles. Conditions: 13 mg mL^−1^ of particles, 1 h, 37 °C, 250 rpm. (**C**,**D**). Release of fucoidan (**C**) and protein (**D**) during incubation of CCF and CCFL microparticles in PBS buffer pH 7.4. Conditions: 13 mg mL^−1^ particles, 37 °C, 250 rpm.

**Figure 9 marinedrugs-23-00428-f009:**
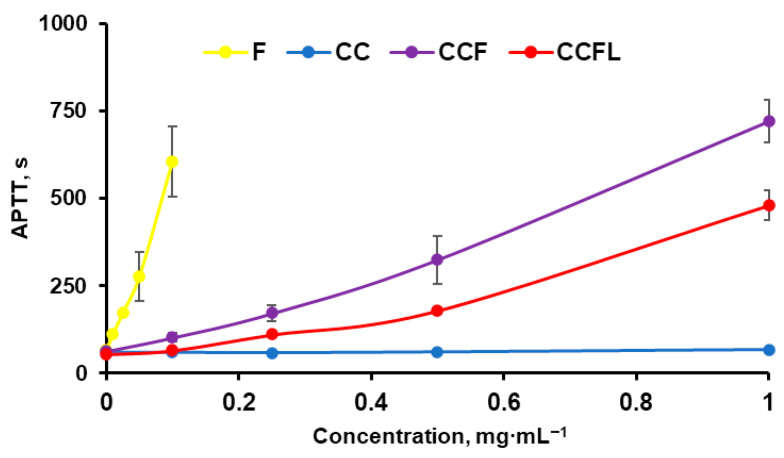
The dependence of activated partial thromboplastin time on the concentrations of fucoidan and microparticles.

**Figure 10 marinedrugs-23-00428-f010:**
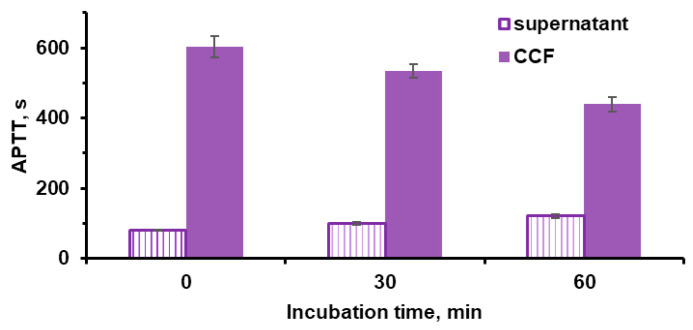
The dependence of the time of the blood plasma coagulation (APTT) induced by CCF microparticles (dark bars) and their supernatants (light bars) on the incubation time. Conditions: 1 mg mL^−1^ CCF, supernatants separated by centrifugation for 2 min at 2000 rpm.

**Figure 11 marinedrugs-23-00428-f011:**
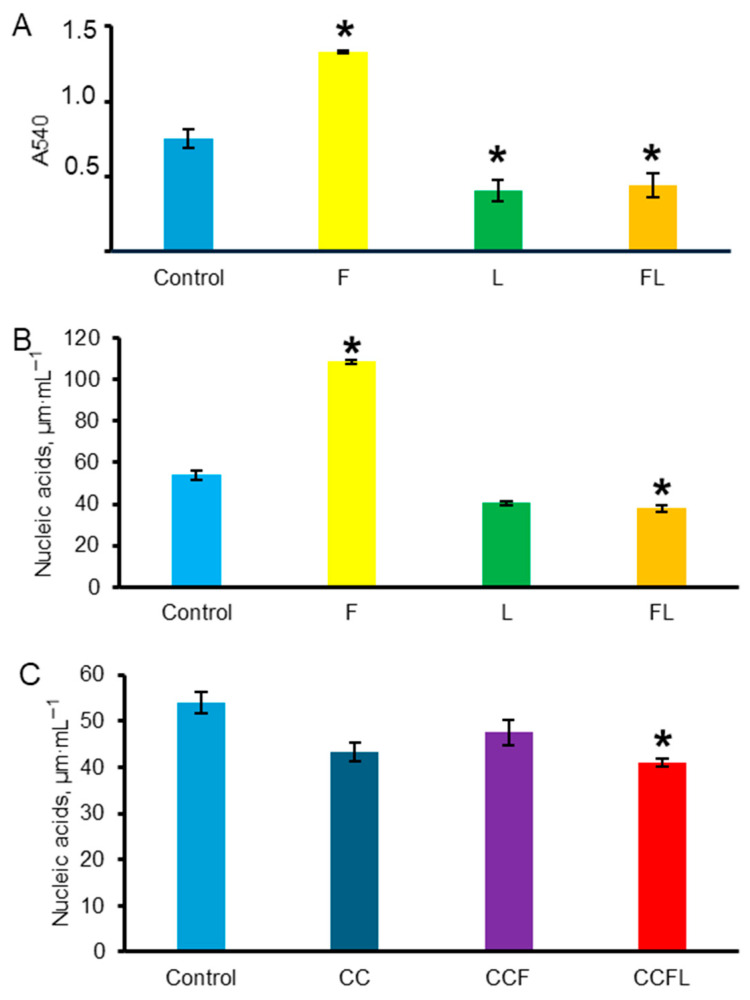
Effects of fucoidan, lactoferrin, and fucoidan–lactoferrin complex, vaterite microparticles (CC), vaterite with fucoidan (CCF), and vaterite with lactoferrin and fucoidan (CCFL) on the growth of *Bacillus subtilis RIK 1285* in M9 medium for 22 h, as evaluated by absorbance at 540 nm (**A**) and by total concentration of nucleic acids (**B**,**C**). Concentration of each biopolymer is 3.125 mg mL^–1^. *, *p* < 0.05 vs. control (without additives) (**A**,**B**). Concentration of the microparticles was 3.125 mg mL^−1^. *, *p* < 0.05 vs. CCF (**C**).

**Table 1 marinedrugs-23-00428-t001:** Physicochemical characteristics of the microparticles and biopolymers.

Sample	D_particle_, µm	Content, %				S, m^2^g^−1^	V, cm^3^g^−1^	D_pores_, nm	ζ-Potential, mV
		Fucoidan		Sulfur	Lactoferrin				
		Determination Method							
		TGA	Dubois	EDS	Lowry				
CC	4.8 ± 1.2			0		12 ± 1	0.04	19.6	−7 ± 2
CCF	4.4 ± 0.9	4.7 ± 0.2	10.3 ± 4.5	1		36 ± 4	0.08	10.0	−11 ± 1
CCFL	3.2 ± 0.5		10.1 ± 1.3	0.7	2.7 ± 0.3	35 ± 4	0.05	5.4	−12 ± 1
Fucoidan		100							−56 ± 1
Lactoferrin					100				10 ± 1
Complex FL									−48 ± 1

**Table 2 marinedrugs-23-00428-t002:** Conditions for microparticle preparation.

Particles	Volume, mL				
	0.1 M Tris-HCl	10 mg mL^−1^ Fucoidan	50 mg mL^−1^ Lactoferrin	1 M CaCl_2_	1 M Na_2_CO_3_
CC	9.0	-	-	3.0	3.0
CCF	1.5	7.5	-	3.0	3.0
CCFL	-	2.5	1.5	3.0	3.0

## Data Availability

The original contributions presented in this study are included in the article/Appendix A. Further inquiries can be directed to the corresponding author.

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
