# Peer review of "Vaterite/Fucoidan Hybrid Microparticles: Fabrication, Loading of Lactoferrin, Structural Characteristics and Functional Properties"

_marinedrugs, 2025, doi:10.3390/md23110428_

Round 1

Reviewer 1 Report

Comments and Suggestions for Authors

Dear Editor,

In this work, authors investigated the properties of hybrid vaterite microparticles with fucoidan obtained from Fucus vesiculosus by co-precipitation and loaded with recombinant human lactoferrin from goats. Physicochemical and in vitro parameters evaluated.  

 Abstract:  ‘creation’ of drug carriers, creation verb is not fit here, change it and check the manuscript for such type of non-scientific verb and replace with suitable scientific verb.

The hybrid microparticles had a smaller diameter, larger surface area, smaller pore size, and more negative ζ-potential than the control vaterite microparticles. Add the values of each parameter.

 The antibacterial properties of hybrid microparticles with fucoidan and lactoferrin against Bacillus subtilis were demonstrated. What the meaning is of ‘demonstrated’?

Introduction: Nanoparticles and microparticles, films, hydrogels, and liposomes have been obtained using fucoidan [12, 20-67 22]. Add the references individually for each preparation.

 -good solubility, which solvents? Add the pharmacokinetics and physicochemical properties of fucoidan.

Fucus vesiculosus, will be in italics.

Results and discussion: Figure 9. Release of fucoidan (A) and (B), why so much sampling time interval difference in release? After 25h directly after 150h sampling. Kept very less sampling time intervals, why?

Why did not perform the TGA and XRD for CCFL and fucoidan? How can confirm the degradation or loading pattern of these without these data?

Need to discuss results more rigor.

Methodology: Table 2. Conditions for microparticle preparation, what is the role of NaCl? Vaterite is a polymorph of calcium carbonate? How calcium carbonate will form by using NaCl?

Comments on the Quality of English Language

Check the manuscript for grammar and sentences errors.

Author Response

We are grateful to the reviewer for the thoughtful comments and feedback. The authors have carefully considered the comments and tried our best to address every one of them.

  1. Abstract:

“Creation’ of drug carriers, creation verb is not fit here, change it and check the manuscript for such type of non-scientific verb and replace with suitable scientific verb”.

We have changed “created” with “development”, “microfabrication”, “manufacturing”, although there is a number of publication with use of “creation” in the same context  (https://link.springer.com/article/10.1023/A:1011442024658#citeas).

The hybrid microparticles had a smaller diameter, larger surface area, smaller pore size, and more negative ζ-potential than the control vaterite microparticles. Add the values of each parameter.

These values are represented in the Table 1 and we really cannot add them to the abstract because of the word count restrictions (200 words maximum).

  The antibacterial properties of hybrid microparticles with fucoidan and lactoferrin against Bacillus subtilis were demonstrated. What the meaning is of ‘demonstrated’?

We have replaced “demonstrated” with “characterized”

  1. Introduction:

Nanoparticles and microparticles, films, hydrogels, and liposomes have been obtained using fucoidan [12, 20-22]. Add the references individually for each preparation.

The lines 67-68 were changed as follows:

“Nanoparticles [12,20] and microparticles [12], films [21], hydrogels [20], and liposomes [22] have been obtained using fucoidan”.

In “References” the lines 588-593 were changed to:

  1. Cunha, L.; Grenha, A. Sulfated seaweed polysaccharides as multifunctional materials in drug delivery applications. Mar. Drugs. 2016, 14(3), 42. doi: 10.3390/md14030042
  2. Liu, W.; Xu, X.; Duan, L.; Xie, X.; Zeng, X.; Huang, Y.; Li, Y.; Zhi. Z.; Pang, J.; Wu, C. Fucoidan-based delivery systems: From fabrication strategies to applications. Food Res. Int. 2025,218, 116752. doi: 10.1016/j.foodres.2025.116752.
  3. Iqbal, M.W.; Riaz, T.; Mahmood, S.; Bilal, M.; Manzoor, M.F.; Qamar, S.A.; Qi X. Fucoidan-based nanomaterial and its multifunctional role for pharmaceutical and biomedical applications. Crit. Rev. Food Sci. Nutr. 2024, 64(2), 354-380. doi: 10.1080/10408398.2022.2106182
  4. Obiedallah, M.M.; Melekhin, V.V.; Menzorova, Y.A.; Bulya, E.T.; Minin, A.S.; Mironov, M.A. Fucoidan coated liposomes loaded with novel antituberculosis agent: preparation, evaluation, and cytotoxicity study. Pharm. Dev. Technol. 2024, 29(4), 311-321. doi: 0.1080/10837450.2024.2332454».

 -good solubility, which solvents? Add the pharmacokinetics and physicochemical properties of fucoidan.

Line 80: “solubility” was changed to “water-solubility”:

“Due to the negatively charged sulphate groups, good water-solubility and high charge density over a wide pH range, fucoidan is capable of forming stable complexes with positively charged proteins [13-18]”.

We consider anionic nature of fucoidan as one of very important physicochemical properties since polyelectrolyte complexes with other biopolymers (proteins, polysaccharides) could facilitate their inclusion into particles. Thus, fucoidan plays a dual role – as a structural component and as an active substance. The studies on the pharmacokinetics of fucoidan require dosage of mg/kg and higher. It is difficult to provide such dosage using vaterite microparticles with fucoidan. We suppose that this formulation will work better via effects on mucosal microenvironment. That’s why we do not discuss pharmacokinetics here, for now.

Fucus vesiculosus, will be in italics.

 It was corrected.

Results and discussion: 

Figure 9. Release of fucoidan (A) and (B), why so much sampling time interval difference in release? After 25 h directly after 150 h sampling. Kept very less sampling time intervals, why?

When the kinetics of fucoidan release from CCF and CCFL microparticles and that of lactoferrin from CCFL was studied we have chosen the time interval so that it corresponded to the possible transit time through the small intestine, this is less than 24 h. The value for 168 h time-point was represented only to show the microparticles’ stability under steady-state conditions (without medium renewal).

The following corrections were added to 3.4 section: 

“For release kinetics, 20 mg microparticles were suspended in 1.5 mL PBS (pH 7.4) and incubated at 37°C (220 rpm). Samples were taken at 10 min, 1, 3, 6, 24 (which corresponded to the possible transit time through the small intestine), and 168 h (to show the microparticles’ stability over a long time period) and analyzed by light microscopy or after centrifugation (2000 rpm, 2 min) for lactoferrin content according to the Lowry method [63] and fucoidan content according to the Dubois method [64]”.

Why did not perform the TGA and XRD for CCFL and fucoidan? How can confirm the degradation or loading pattern of these without these data?

Since fucoidan does not include polymorphs of calcium carbonate, we did not perform XRD for it. XRD of CCFL was not performed because of very similar CCF and CCFL morphology according to SEM (Figure 3).

Analyzing CCF by thermogravimetry (TGA, Figure S1), we compared our results with the data for the similar fucoidan (from Fucus vesiculosus (Sigma-Aldrich)) published in [Flórez-Fernández N, Pontes JF, Guerreiro F, Afonso IT, Lollo G, Torres MD, Domínguez H, Costa AMRD, Grenha A. Fucoidan from Fucus vesiculosus: Evaluation of the Impact of the Sulphate Content on Nanoparticle Production and Cell Toxicity. Mar Drugs. 2023, 7, 21(2): 115. doi: 10.3390/md21020115]:

Fig. from [63]. Thermogravimetric analysis diagram of fucoidan from F. vesiculosus. Green (continuous) lines refer to the variation in the mass of the polymers throughout the process (TG/%), and pink (dashed) lines are the respective derivatives (DTG/(%/min)) [Flórez-Fernández N., 2023]

For a similar fucoidan, an initial change in sample mass of 12% was recorded, associated with water evaporation, followed by a more intense change at 218°C, which corresponds to the degradation of the biopolymer. The given temperature ranges and comparison with the TGA curve for CC microparticles were used to determine the fucoidan content of CCF microparticles. The change in the mass of CC and CCF particles at temperatures above 559°C was observed largely due to the decomposition of calcium carbonate. The observed thermal characteristics of the CCF microparticles indicate that fucoidan is stable within them at room temperature. 

We did not conduct TGA analysis of fucoidan, lactoferrin, their complex, or CCLF particles, as the degradation of the lactoferrin glycoprotein and sulfopolysaccharide must occur within a similar temperature range. Therefore, it could be difficult to separate the fucoidan and lactoferrin content in the particles using TGA.

To analyze the composition of hybrid particles with fucoidan CCF and CCFL, we proposed dissolving the CaCO3 matrix with hydrochloric acid and determining the content of components using known spectrophotometric methods. Fucoidan content was analyzed by the Dubois phenol-sulfur method using a calibration curve for the polysaccharide and glycoprotein lactoferrin at a wavelength of 485 nm (Fig. S2). The optical absorbance of lactoferrin in the Dubois method was approximately six times lower than that of fucoidan. The fucoidan content in CCF particles, determined by the Dubois method, was higher than that determined using the TGA method (4.7 ± 0.2 and 10.3 ± 4.5, respectively).

Need to discuss results more rigor.

Calibration dependences for fucoidan and lactoferrin according to the data of Dubois assay are represented in Figure S2 (A, B). Reference to the publication [Flórez et al 2023] was added to the 3.3.5 section and to the References as [63]:

3.3.5. Thermogravimetric analysis

Thermogravimetric analysis (TGA) was conducted on an SDT Q600 analyzer (Thermo Fisher Scientific, TA Instruments). Samples were heated from 30 to 1000°C at 10°C min-1 under an air flow (100 mL min-1). For the processing and interpretation of TGA data for CCF, the TGA data for fucoidan from F. vesiculosus [63] was used.  

Methodology: Table 2. Conditions for microparticle preparation, what is the role of NaCl? Vaterite is a polymorph of calcium carbonate? How calcium carbonate will form by using NaCl?

In the Table 2 “NaCl” was changed with «Na2CO3».

Reviewer 2 Report

Comments and Suggestions for Authors

The introduction is very well written and detailed; however, it is rather lengthy, which distracts the reader from the key information. Its length should be reduced by approximately half.

Figure 3: In the diameter distribution diagrams, the sizes on the X-axis cannot be clearly distinguished due to the tick labels currently used.

Line 370: The methodology followed or the research group should be mentioned (e.g., “according to…”).

Line 374: The conditions and duration of lyophilization should be stated.

Figure S2: The typographical error in the figure caption should be corrected.

Section 3.2: The methods used are not sufficiently explained; providing only a reference is not adequate. They should be described in detail, including the instrumentation used (e.g., Dubois method, Lowry method). Additionally, the description “In a beaker with a bottom diameter of 45 mm under magnetic stirring (200 rpm), the required volumes of solutions were sequentially added as specified in Table 2” is not sufficient for a scientific paper. More detailed procedural information should be provided, such as the time intervals between the addition of each solution and the total mixing time before the microparticles were collected.

Section 2.1 (Lines 170–176): This paragraph belongs in the Materials and Methods section, specifically under Section 3.2.

Regarding the TGA method, it should be clarified how the authors demonstrate that the observed mass loss is solely due to the fucoidan content. The TGA thermograms of all raw materials should be provided to allow for proper comparison.

Author Response

We are grateful to the reviewer for the thoughtful comments and feedback. The authors have carefully considered the comments and tried our best to address every one of them.

The introduction is very well written and detailed; however, it is rather lengthy, which distracts the reader from the key information. Its length should be reduced by approximately half.

We agree with the reviewer that the introduction is long, but it allows readers to become familiar with all the problems associated with the need to create hybrid vaterite microparticles based on fucoidan and the inclusion of lactoferrin in them.

Figure 3: In the diameter distribution diagrams, the sizes on the X-axis cannot be clearly distinguished due to the tick labels currently used.

The figure was corrected.

Figure 3. SEM and diameter distribution diagram of microparticles: control CC (A), hybrid with fucoidan CCF (B), and hybrid microparticles with fucoidan and lactoferrin CCFL (C).

Line 370: The methodology followed or the research group should be mentioned (e.g., “according to…”).

Added:  "according to the method developed in the group of N. Balabushevich [41]” (line 379)

Line 386: The conditions and duration of lyophilization should be stated.

The text was changed as follows: “After synthesis, the microparticles were centrifuged at 1000 rpm for 1 min, washed twice with double-distilled water, frozen, lyophilized (freezing at –400C, drying for 24 h) and weighed”.

Figure S2: The typographical error in the figure caption should be corrected.

It was corrected:

Figure S4. X-ray phase diagrams of vaterite microparticles.

Section 3.2: The methods used are not sufficiently explained; providing only a reference is not adequate. They should be described in detail, including the instrumentation used (e.g., Dubois method, Lowry method). Additionally, the description “In a beaker with a bottom diameter of 45 mm under magnetic stirring (200 rpm), the required volumes of solutions were sequentially added as specified in Table 2” is not sufficient for a scientific paper. More detailed procedural information should be provided, such as the time intervals between the addition of each solution and the total mixing time before the microparticles were collected.

The Section 3.2 was extended with the following information (in yellow):

The synthesis of control vaterite microparticles (CC), hybrid microparticles with fucoidan (CCF), and complex hybrid microparticles with fucoidan and lactoferrin (CCFL) were performed according to [41]. In a beaker with a bottom diameter of 45 mm under magnetic stirring (200 rpm), the required volumes of solutions were sequentially added as specified in Table 2. For ССF microparticles, 3 mL of 1 M CaCl2, 7.5 mL 10 mg mL 1 of fucoidan in 0.1 M Tris buffer, 1.5 mL water was stirred for 10 min followed by the addition of 3 mL of 1 M Na2CO3. Stirring was continued for 45 s, then the suspension was left for 15 min without stirring for precipitation. CC microcrystals were fabricated following the same procedure without addition of polysaccharide into the CaCl2 solution. For CCFL microparticles, fucoidan and lactoferrin were pre-incubated for 10 min to form a complex. After synthesis, the microparticles were centrifuged at 1000 rpm for 1 min, washed twice with double-distilled water, frozen, lyophilized, and weighed.

Detailed description of Dubois method and Lowry method was added to Supplementary section as follows:

  1. The colorimetric method for determination of sugars (Dubois Assay)

The standard fucoidan concentrations 0 - 0.35 mg mL-1 were used for calibration of the method (Figure S2 (A)) by Dubois assay as follows: 0.15 mL fucoidan solution or negative control was mixed with 0.15 mL 5% phenol and 0.75 mL concentrated sulfuric acid. The mixture was vortexed and after 30 min, the absorbance at 485 nm was read. Microparticles CCF or CCFL were first resuspended in double-distilled water (1 mg mL-1) and then 0.15 mL suspension was mixed with other reagents as described above.

As Figure S2 (B) shows, oligosaccharides in glycoprotein lactoferrin also can be determined by Dubois assay but 0 – 0.5 mg mL-1 lactoferrin absorbance was 6 times less than that of fucoidan.

Figure S2. Calibration dependence for concentration of fucoidan (A) and lactoferrin (B) determined by Dubois assay.

  1. Lowry’s protein assay

The following reagents were prepared: solution A was 2% (w/v) sodium carbonate in 0.1 M sodium hydroxide; solution B was 0.781 g CuSO4 . 5H2O and 1.3837g sodium citrate dissolved in 100 mL H2O. Solution C was the mixture of 50 volume of solution A and 1 volume of solution B prepared before determination. Folin-Ciocalteau reagent was diluted to 1 M acid according to the supplier’s instruction.

To 0.2 mL of the 0-0.5 mg mL-1 lactoferrin solution, 1 mL of solution C was added, mixed thoroughly by vortexing and was left at room temperature for 10 min. Then 0.1 mL of diluted Folin-Ciocalteau reagent was added, mixed rapidly, and incubated for 30 min at room temp. Absorbance at 750 nm was measured against reagent blank not containing protein. The calibration dependence (Figure S3) was used to determine lactoferrin concentration in the experimental samples.

Figure S3. Calibration dependence for determination of lactoferrin by Lowry’s method.

The lactoferrin incorporation into the microparticles was calculated from the difference between its concentration in the initial solution and in supernatants and washing solutions at particles fabrication. To determine protein in the microparticles, they were previously dissolved in 15% HCl.

Section 2.1 (Lines 170–176): This paragraph belongs in the Materials and Methods section, specifically under Section 3.2.

We agree that it looks rather like “Materials and Methods” (section 3.2) but we consider important to place some part of the method also here since it helps to understand better the results. We added this indication to the text:

«To determine the optimal strategy for incorporating lactoferrin (L, 80 kDa; pI 8.0–8.5) into microparticles, two loading methods were compared: co-precipitation and adsorption (Fig. 2). As described below in the “Materials and Methods” section, lactoferrin adsorption was performed on dried CCF particles to obtain CCF-L microparticles. For co-precipitation, lactoferrin and fucoidan were pre-incubated in a 1 : 1 mass ratio to form the FL polyelectrolyte complex, the existence of which we had previously confirmed [58], and then quickly mixed with calcium chloride and sodium carbonate solutions to obtain CCFL microparticles. »

Regarding the TGA method, it should be clarified how the authors demonstrate that the observed mass loss is solely due to the fucoidan content. The TGA thermograms of all raw materials should be provided to allow for proper comparison.

Analyzing CCF by thermogravimetry (TGA, Figure S1), we compared our results with the data for the similar fucoidan (from Fucus vesiculosus (Sigma-Aldrich)) published in [Flórez-Fernández N, Pontes JF, Guerreiro F, Afonso IT, Lollo G, Torres MD, Domínguez H, Costa AMRD, Grenha A. Fucoidan from Fucus vesiculosus: Evaluation of the Impact of the Sulphate Content on Nanoparticle Production and Cell Toxicity. Mar Drugs. 2023, 7; 21(2):115. doi: 10.3390/md21020115]:

Fig. from [63]. Thermogravimetric analysis diagram of fucoidan from F. vesiculosus. Green (continuous) lines refer to the variation in the mass of the polymers throughout the process (TG/%), and pink (dashed) lines are the respective derivatives (DTG/(%/min)) [Flórez-Fernández N., 2023].
